# Identifying Endogenous Proteins of Perennial Ryegrass (*Lolium perenne*) with *Ex Vivo* Antioxidant Activity

**DOI:** 10.3390/proteomes13010008

**Published:** 2025-02-05

**Authors:** Kathrine Danner Aakjær Pedersen, Line Thopholm Andersen, Mads Heiselberg, Camilla Agerskov Brigsted, Freja Lyngs Støvring, Louise Mailund Mikkelsen, Sofie Albrekt Hansen, Christian Enrico Rusbjerg-Weberskov, Mette Lübeck, Simon Gregersen Echers

**Affiliations:** Department of Chemistry & Bioscience, Aalborg University, Fredrik Bajers Vej 7H, DK-9220 Aalborg, Denmark; kdap20@student.aau.dk (K.D.A.P.); ltan19@student.aau.dk (L.T.A.); mheise19@student.aau.dk (M.H.); cbrigs19@student.aau.dk (C.A.B.); fstavr19@student.aau.dk (F.L.S.); sofieah@bio.aau.dk (S.A.H.); cerw@bio.aau.dk (C.E.R.-W.); mel@bio.aau.dk (M.L.)

**Keywords:** perennial ryegrass, green biorefining, wet fractionation, antioxidant activity, size fractionation, bottom-up proteomics, bioinformatics, enrichment analysis

## Abstract

**Background**: During the initial steps of green biorefining aimed at protein recovery, endogenous proteins and enzymes, along with, e.g., phytochemical constituents, are decompartmentalized into a green juice. This creates a highly dynamic environment prone to a plethora of reactions including oxidative protein modification and deterioration. Obtaining a fundamental understanding of the enzymes capable of exerting antioxidant activity *ex vivo* could help mitigate these reactions for improved product quality. **Methods**: In this study, we investigated perennial ryegrass (*Lolium perenne* var. Abosan 1), one of the most widely used turf and forage grasses, as a model system. Using size exclusion chromatography, we fractionated the green juice to investigate *in vitro* antioxidant properties and coupled this with quantitative bottom-up proteomics, GO-term analysis, and fraction-based enrichment. **Results**: Our findings revealed that several enzymes, such as superoxide dismutase and peroxiredoxin proteoforms, already known for their involvement in *in vivo* oxidative protection, are enriched in fractions displaying increased *in vitro* antioxidant activity, indicating retained activity *ex vivo*. Moreover, this study provides the most detailed characterization of the *L. perenne* proteome today and delivers new insights into protein-level partitioning during wet fractionation. **Conclusions**: Ultimately, this work contributes to a better understanding of the first steps of green biorefining and provides the basis for process optimization.

## 1. Introduction

Changes in the global climate have become a planetary crisis, drawing considerable attention, and need to be addressed through a green transition. According to the UN’s Sustainable Development Goals, this involves implementing emission-free energy systems, sustainable production practices, and effective by-product and waste management to mitigate environmental degradation, ensure long-term sustainability, and meet the needs of a growing human population. Plant-based foods represent a significant component of this transition. The production of plant-based foods is generally less resource-intensive and environmentally destructive, as it typically requires less land, water, and energy compared to raising livestock and shows significantly lower levels of greenhouse gas emissions [1]. Conventional animal-based products are typically the primary sources of protein in traditional diets, particularly in Western countries. Given that proteins are essential macronutrients for human nutrition, it becomes necessary to prioritize the development of sustainable plant-based protein sources that can effectively supplement or replace traditional sources [2]. This especially applies to plant-based protein ingredients, as they provide a concentrated source of protein, enabling the creation of a wide range of products allowing for diverse dietary options and specific nutritional needs.

In recent years, green leaves such as alfalfa, clover, grasses, immature cereals, and plant shoots have gained increasing interest as promising protein sources [3,4]. Grasses, in particular, are versatile, thrive in various climates and soils from agricultural to marginal lands, and cover around 70% of the global land area [5]. Generally, grasses contain high amounts of protein, ranging between 6 and 26% in dry matter, with protein levels being linked to the growth stage of the grass and affected by soil nutrition [5,6]. This makes grasses a ubiquitous resource for protein production.

Grasses are especially favored in temperate climates, which provide adequate soil temperatures, sunlight, and rainfall throughout the year [5,7]. Perennial ryegrass (*Lolium perenne*) is a common indigenous temperate crop and one of the most widely used turf and forage grasses [8]. Compared to annual crops, perennials, such as *L. perenne*, can achieve greater biomass production due to their extended growing period, which allows for the enhanced interception of solar radiation and repeated harvest [9]. Additionally, the deeper root systems of perennial crops facilitate access to water and nutrients from deeper soil layers, which contributes to improved resilience against environmental stresses such as drought. This also supports better nutrient uptake and reduces nitrate leaching associated with fertilization [9]. During the vegetative growth phase and with high nitrogen input, the protein content of ryegrass species can exceed 30% in dry matter [10].

Utilizing grasses for protein production in temperate regions like Northern Europe can reduce dependence on imported plant proteins, such as soy, which require warmer climates for cultivation, thereby minimizing the environmental footprint associated with transport. Furthermore, locally produced plant-based proteins can strengthen the local supply chain and contribute to long-term sustainable food security. Currently, most of the market for grass is limited to feed, primarily for ruminants, as the direct use of the biomass is unsuitable for monogastric animals and humans [9,11]. However, through biorefinery processes, it is possible to extract the proteins from the grass, making it available for both monogastric and human consumption. Green biorefining is the use of fresh, green biomass as feedstock for the production of various products, such as food, feed, chemicals, bioactive components, energy, and/or biofuels. Strategies aimed at utilizing all side streams and optimizing production are generally employed [6,12]. So far, most studies focus on green biomass from mixed fields, where perennial grasses are combined with primarily clover species but also other green legume crops [5,13]. Mixed species have several advantages in terms of, e.g., increased nitrogen fixation and more stable biomass yields. Consequently, studies with an exclusive focus on *L. perenne* are limited. Nevertheless, the potential for the biorefining of perennial grasses has been highlighted for more than a decade [14,15,16,17]. Regardless of origin, the first step of the process entails extracting protein by the disruption of the plant tissue to release the intracellular content. This is often referred to as wet fractionation and is usually performed by mechanically fractionating using a press, resulting in a green juice and a fibrous pulp [18]. The pulp itself has direct applicability and its use for the production of cellulose and hemicellulose fibers [19,20] and as feed for dairy cows [21] has been demonstrated specifically for *L. perenne*. The green juice is rich in proteins, chlorophyll, membrane fragments, and other cellular compounds and debris. Secondary processing can be performed to separate and concentrate the proteins further [22,23]. The green juice from *L. perenne* has been used for the production of, e.g., protein-rich pig feed [24]. The production of a food-grade protein product from *L. perenne* green juice is still in the very early stages but interest is increasing [25,26].

As wet fractionation disrupts cell compartments, the extracted proteins consist of both soluble cytoplasmic proteins and insoluble membrane proteins. The cytoplasmic proteins, typically referred to as “white protein”, include mainly chloroplast enzymes. These are soluble, odorless, and tasteless proteins with a clear color, making them suitable for food applications [18]. One of the most abundant and widely studied proteins in this relation is the photosynthetic enzyme ribulose-1,5-bisphosphate carboxylase/oxygenase (RuBisCO), which possesses great nutritional value and functional properties such as foaming, gelling, and emulsification [25,26,27]. Under optimal conditions, RuBisCO constitutes about 50% of the soluble protein of plant leaves by mass; however, several other functional proteins could be present in the green juice, indicating the potential of green biomass as a multifunctional protein source for food applications.

Unfortunately, several challenges arise during processing in green biorefineries, which can result in undesirable changes in protein properties and functionality. For instance, proteolysis, aggregation, and chemical modifications occur as soluble proteins, enzymes, and phytochemicals are mixed *ex vivo* after being removed from their original subcellular location and environment. Additionally, homogenization operations, exposure to UV light, or the release of complex-bound transition metals, can facilitate, e.g., amino acid side chain modification and protein cross-linkage through oxidative processes [28,29]. Consequently, antioxidant capacity is necessary to maintain the native state and functionality of the proteins in the green juice.

Plants are typically considered abundant sources of natural antioxidant metabolites. Hitherto, research on plant antioxidants in relation to food applications has predominantly focused on non-enzymatic compounds, such as polyphenols, flavonoids, vitamins, and volatile chemicals, rather than proteins [30,31]. When plant proteins are studied for their antioxidant properties, the focus is often on protein-derived peptides [32,33,34,35,36,37]. However, during green biorefining, the primary goal is often to maximize protein concentration as much as possible, which includes the removal of antioxidant metabolites and peptides. Moreover, biorefining ideally aims to preserve proteins in their native form, thereby maintaining solubility and original functionality. This highlights the importance of exploring the antioxidant potential of whole plant proteins. A deeper understanding of this potential would allow the optimization of enrichment processes, ultimately enhancing product stability, extending shelf-life, and broadening their applicability as food ingredients.

The research on antioxidant proteins has mainly been focused on the intricated redox balance of cells *in vivo*, where antioxidants are highly interactive, constituting a complex defense system against redox imbalance. In relation to *L. perenne*, the expression level during stress conditions of known enzymatic antioxidants such as ferredoxin–thioredoxin reductase (FTR), thioredoxin (Trx), superoxide dismutase (SOD), catalase (Cat), ascorbate peroxidase (APX), glutathione reductase (GR), monodehydroascorbate reductase (MD(H)AR), and dehydroascorbate reductase (DHAR), have been investigated to improve agricultural productivity [38,39,40,41]. As eukaryotic cells are highly compartmentalized, the subcellular location of proteins may be crucial for protein function. Hereby, their catalytic activity is influenced by cellular dynamics and variation in compartmental conditions [42]. To our knowledge, no prior studies have investigated the *ex vivo* antioxidant activity of native proteins from *L. perenne* green juice. Therefore, less is known about the activity of these known antioxidant proteins when separated from cellular conditions such as the presence of co-factors and synergistic effectors.

This knowledge gap provides an opportunity to uncover novel insights at a fundamental level of native antioxidant proteins, as inherent antioxidants could enhance their value in relation to food applications. In this study, the *ex vivo* antioxidant activity of fractionated *L. perenne* green juice protein is investigated and responsible effectors are evaluated. Moreover, the study provides the first comprehensive proteomic analysis of *L. perenne* and provides insights into the fate of individual proteins during the first step of a biorefinery process.

## 2. Materials and Methods

### 2.1. Grass Cultivation and Wet Fractionation

*Lolium perenne* (var. Abosan 1, DLF Seeds A/S, Roskilde, Denmark) was cultivated in two different batches. Seeds were planted in seed boxes with Forest Gold, pH 6, soil (Pindstup, Ryomgaard, Denmark) consisting of 70% sphagnum and 30% wood fiber soil and placed in a FITOCLIMA 5.000 PLH (Aralab, Rio de Mouro, Portugal) climate chamber (18 h light/6 h dark, 195 μmol·m^−2^·s^−1^, 21/19 °C, 50% relative humidity). Batches 1 and 2 were cultivated for 48 and 50 days, respectively, and the soil was kept moist during the cultivation. Both batches were wet fractionated immediately after harvest using an Angelia 8500 S juicer (Angel Co., Ltd., Naarden, Netherland) at standard settings. Batch 1 juice was used for protein separation (protein fractions), while batch 2 was used for the analysis of the raw grass, pulp, and juice (initial crude fractions).

### 2.2. Crude Protein Estimation and Dry Matter Analysis

Crude protein was estimated by elemental analysis using a FlashSmart™ CHNS/O (Thermo Fisher Scientific, Waltham, MA, USA). Lyophilized grass, pulp, and juice were cryogenically ground, and 2–3 mg was packed in soft tin capsules. Acetanilide (OAE Labs, Exeter, UK) was used as a reference standard for calibration, and analyses were run on CN mode with helium as the carrier gas at a flow rate of 140 mL/min. The combustion furnace temperature was 950 °C and the detector oven temperature was 50 °C. Crude protein content was calculated from the nitrogen content by using the nitrogen-to-protein conversion factor of 6.25. All samples were analyzed as triplicates.

The dry matter content (DM) of grass and pulp was determined by drying 2–5 g of biomass at 105 °C overnight. The DM of the juice was determined using 20 µL and drying it overnight at 60 °C. All samples were analyzed with at least three replications.

### 2.3. One-Dimensional SDS-PAGE

One-dimensional SDS-PAGE was performed under both reducing and non-reducing conditions using SurePAGE 4–20% polyacrylamide gels (Genscript, Picastaway, NJ, USA) and a Tris-MOPS SDS Running Buffer system (Genscript, Picastaway, NJ, USA). Liquid samples of protein fractions were mixed with SDS sample buffer (50 mM Tris pH 6.8, 2% SDS, 10% glycerol, 0.02% bromophenol blue, 12.4 mM EDTA, and, for reducing gels, 1 M DTT) and ddH_2_O in a 10:7:5 ratio and subsequently incubated for 10 min at 95 °C and 200 rpm. For solid samples (crude fractions), lyophilized biomass was incubated in sample buffer for 10 min at 95 °C and centrifuged and the supernatant was recovered for SDS-PAGE analysis. The amount of dry biomass and the volume of liquid fractions were standardized according to crude protein estimation and Qubit concentration (see below), respectively, to reach 18 μg protein in the wells. The electrophoreses were carried out at 160 V for 40 to 60 min until the dye front reached the bottom of the gel and was subsequently stained overnight with InstantStain Coomassie blue (Kem-En-Tec Nordic A/S, Taastrup, Denmark) before imaging using a ChemDoc MP imaging System (BioRad, Hercules, CA, USA).

### 2.4. Protein Fractionation by Size Exclusion Chromatography

The juice from batch 1 was centrifuged at 3134 rcf for 20 min, and the supernatant was immediately transferred to new tubes and centrifuged at 14,100 rcf for 10 min. Subsequently, the supernatant was filtered through a 0.45 μm LABSOLUTE sterile filter (TH. Geyer, Ballerup, Denmark). The green juice was fractionated by size exclusion chromatography (SEC) using an NGC chromatography system (Bio-Rad, Hercules, CA, USA) equipped with a 320 mL HiPrep^TM^ 26/60 with Sephacryl^®^ S-200 (dextran-acrylamide copolymer) HR column (Cytiva, Marlborough, MA, USA). The column was washed and subsequently equilibrated using 0.2 M phosphate buffer (pH 6.5) according to the manufacturer’s guidelines. Ten milliliters of the centrifuged and filtered green juice was injected into the system. Using a flow rate of 1.8 mL/min, fractions of 4 mL were collected while absorbance at 280 nm and conductivity were continuously measured. The protein concentration and the 260/280 ratio were subsequently measured by UV spectroscopy using an SDS-11 FX (Denovix, Wilmington, DE, USA) at standard settings (1 Abs = 1 mg/mL). All protein fractions were kept cool at pH 6.5 throughout the rest of the experimental work.

#### Size Estimation of Protein Fractions

To generate a calibration curve, standards (all Cytiva, Marlborough, MA, USA) including 1 mL Blue Dextran 2000 (2 mg/mL in MilliQ) and a calibration mix consisting of ovalbumin (4 mg/mL in PBS), conalbumin (3 mg/mL in PBS), and aldolase (4 mg/mL in PBS) was injected and monitored at 280 nm with a flow rate of 1.2 mL/min. The distribution coefficient (*K_d_*) for the standards was calculated as:$$K_{d}=\frac{V_{e}-V_{0}}{V_{c}-V_{0}},$$
where *V_e_* is the elution volume, *V*_0_ is the void volume, and *V_c_* is the column volume. The calibration curve was constructed by plotting *K_d_* against the logarithmic molecular weight (MW) of the standard proteins. By linear regression, the MW can be estimated for the fractions based on the average elution volume of each fraction.

### 2.5. DPPH Radical Scavenging Activity Screening and Fraction Selection

Each SEC fraction was screened for radical scavenging activity (RSA) through the DPPH assay, according to Nicklisch and Waite [43], with minor adjustments. Briefly, the assay was performed in 96-well microplates (flat base and transparent, Sardstedt, Germany) using a working volume of 150 μL. The protein fractions were standardized in volume and subjected to a two-fold tree-point dilution series using phosphate buffer. Subsequently, Triton X-100 (AppliChem, Darmstadt, Germany) and methanolic DPPH (Sigma Aldrich, Søborg, Denmark) were added to reach final concentrations of 0.3% (*v*/*v*) and 300 μM, respectively. Trolox (TCI EUROPE N.V., Zwijndrecht, Belgium) was used as a positive control and PBS as a negative control. Furthermore, blanks of Trolox or protein fraction, phosphate buffer, and Triton X-100 were made to determine and account for background absorbance. The microplates were incubated for 1 h (darkness, ambient temperatures, 200 rpm), and subsequently, absorbance was measured at 517 nm on a Spark microplate reader (Tecan, Männedorf, Switzerland). The absorbance was converted into a percentage of RSA by the following equation:$$Activity\left(\%\right)=\left(1-\frac{A_{S}-A_{B}}{A_{C}}\right)\cdot 100\%$$
where A_S_ is absorbance measured for the sample, A_B_ is absorbance for the blanks, and A_C_ is absorbance for the negative control. All samples were analyzed as a three-point dilution series. Based on empirical observations, high DPPH radical scavenging, and a desire to cover different parts of the MW range, 15 fractions were selected for further downstream analyses.

### 2.6. Quantification of Antioxidant Properties in Selected Protein Fractions

Prior to further downstream analyses, the selected fractions were purified to remove potential interference from buffer salts and other components, ensuring a better estimate of concentrations.

#### 2.6.1. Desalting and Protein Concentration Determination

The selected fractions were desalted using PD-10 (Sephadex G-25 resin) columns (Cytiva, Marlborough, MA, USA) according to the manufacturer’s guidelines with ddH_2_O as the eluent. The protein concentrations of PD-10 eluates were quantified using the Qubit^TM^ Protein Assay Kit and Qubit^4^ Fluorometer (Thermo-Fisher Scientific, Waltham, MA, USA).

#### 2.6.2. Protein Concentration

To increase protein concentration, approximately 3 mL of each fraction was lyophilized and resuspended in 1.5 mL Milli-Q water. To check that all protein was solubilized, samples were centrifuged at 600 rcf for 5 min. If the protein was not solubilized after two rounds of vortexing and 10 min of sonication, the supernatant was used. Protein concentration in the resuspended samples was measured by Qubit, as previously described. The concentration range used for DPPH radical scavenging was based on results from DPPH screening. As the iron chelating assay was not employed for screening (incompatible with PBS), similar concentrations to the DPPH assay were used.

#### 2.6.3. DPPH Radical Scavenging Activity

The DPPH RSA assay was performed similarly to that described above. Briefly, a working volume of 60 μL was used and a dilution series with a dilution factor of 1.25 was made using Milli-Q water. Trolox was used as a positive control with concentrations ranging from 18.8 to 75 μM and included on all analyzed microplates.

#### 2.6.4. Iron Chelation Activity

The iron chelating activity (ICA) of the protein fractions was investigated according to the method of Sabeena Farvin et al. [44] with minor adjustments. The same initial concentrations of protein fractions, dilution factor, and working volume were used as described for the DPPH RSA assay. FeCl_2_ (Alfa Aesar, Ward Hill, Massachusetts, USA) and Ferrozine (Serva, Heidelberg, Germany) solutions were added to the microplate, reaching final concentrations of 40 μM and 200 μM, respectively. EDTA (AppliChem, Darmstadt, Germany) was used as a positive control with concentrations ranging from 1.6 to 50 μM and included on all analyzed microplates. A negative control without EDTA and a blank without Ferrozine were made following the same principle as for the DPPH assay. The microplates were incubated for 10 min (darkness, ambient temperatures, 200 rpm). Absorbance was measured at 562 nm using a Spark microplate reader (Tecan, Männedorf, Switzerland).

#### 2.6.5. EC50 Calculations

Percentages of RSA and ICA were calculated as described above. The blanks for Trolox and EDTA also served as blanks for all fractions. A linear or logarithmic regression was fitted to the activity against the concentrations to account for the expected sigmoidal shape of the curve. The aim was the best curve fit at 50% activity, where a linear trend is expected. If data points were positioned above 80% activity, a logarithmic curve fit was used. EC50 was calculated depending on a linear or logarithmic model. EC50 values were converted to molar concentrations by using the estimated average MW for each fraction based on the mean SEC retention volume of the respective fraction.

### 2.7. Bottom-Up Proteomics by LC-MS/MS

Quantitative protein identification by bottom-up proteomics (BUP) was performed for both the selected SEC fractions and the initial crude fractions. Due to the different nature of the samples, different sample preparation methods were employed during preparation.

#### 2.7.1. In-Solution Digest of Selected SEC Fractions

A volume of desalted protein fractions, corresponding to a protein mass of 10 μg, was collected and lyophilized overnight for proteomic analysis by MS. The lyophilized fractions were resuspended in 20 μL of digestion buffer (1% (*v*/*v*) sodium deoxycholate (SDC, Sigma-Aldrich, Søborg, Denmark) in 50 mM triethylammonium bicarbonate (TEAB, Merck, Søborg, Denmark)), heated to 99 °C on a heating block for 8 min and then cooled below 37 °C. The samples were incubated for 30 min at 37 °C with 0.8 μg tris(2-carboxyethyl)phosphine hydrochloride (TCEP, Sigma-Aldrich, Søborg, Denmark) before 1 μg iodoacetamide (IAA, Fluka Biochemika, Buchs, Switzerland) was added and incubation continued for another 20 min in the dark. To digest the proteins, 0.2 μg og sequencing-grade trypsin (Promega, Madison, WI, USA) was added and the samples were incubated overnight at 37 °C. To precipitate the SDC, formic acid (FA, VWR, Søborg, Denmark) was added to a final concentration at 0.5% (*v*/*v*) and incubated for 5 min at ambient temperatures. The digests were centrifuged at 11,336 rcf for 20 min at 4 °C. The supernatant from each fraction was collected and further centrifuged for 20 min just before microcolumn purification using in-house prepared C-18 StageTips according to Rappsilber et al. [45]. The purified samples were dried using a vacuum concentrator before resuspension and analysis.

#### 2.7.2. Protein Extraction and In-Solution Digest of Crude Fractions

The initial crude fractions (grass, juice, and pulp) were prepared in triplicates using the iST kit for plant tissue (PreOmics, Planegg, Germany) according to manufacturer guidelines with minor modifications. Briefly, lyophilized biomass corresponding to 70 μg protein (by N ∗ 6.25) was weighed off in LoBind tubes (Eppendorf, Hamburg, Germany), dissolved in 800 μL Lyse buffer, and heated to 95 °C for 5 min at 1000 rpm in a Thermomixer (Eppendorf, Hamburg, Germany). Next, the solution and as much biomass as possible were transferred to a 1 mL AFA tube (Covaris, Wobum, MA, USA) for focused ultrasonication-assisted extraction using an M220 focused ultrasonicator (Covaris, Wobum, MA, USA). For the fibrous fractions (raw grass and pulp), samples were subjected to four cycles of the protein extraction protocol specified by the manufacturer (peak incident power of 75 W, a duty factor of 10%, 200 cycles per burst, and 180 s per cycle at 6 °C), while the more soluble juice fraction was subjected to one cycle. Next, samples were transferred to new LoBind tubes and heated to 95 °C at 1000 rpm for another five minutes. Samples were cooled and followed by the addition of resuspended trypsin/LysC for digestion at 37 °C and 500 rpm for three hours. Afterward, the digests were transferred to iST cartridges for washing (including “Wash 0” for plant tissue), double elution, and drying, as described by the manufacturer.

#### 2.7.3. LC-MS/MS Analysis

The dried protein digests were resuspended in 20 μL of buffer (2% (*v*/*v*) acetonitrile, 0.1% (*v*/*v*) formic acid for protein fractions, and “LC Load” from the iST kit for crude fractions) and the peptide concentration was measured with UV spectroscopy at standard settings (1 Abs = 1 mg/mL) and diluted as required. The buffer served as a blank. Samples were loaded onto a 96-well microplate (Thermo Fisher Scientific, Waltham, Massachusetts, USA). The microplate was covered with sealing tape and loaded into an nLC-1200 ultra-high-performance liquid chromatography system (Thermo Fisher Scientific, Waltham, MA, USA) with ESI coupled to a Q Exactive HF tandem mass spectrometer (Thermo Fisher Scientific, Waltham, MA, USA). Approximately 1 μg of the digest was loaded on a PEPMAP precolumn (75 μm × 2 cm, C18, 3 μm, 100 Å) followed by separation on an analytical PEPMAP column (75 μm × 50 cm, C18, 2 μm, 100 Å). The mobile phase, consisting of solvent A (0.1% (*v*/*v*) formic acid) and solvent B (80% (*v*/*v*) acetonitrile, 0.1% (*v*/*v*) formic acid), was run with a stepwise gradient from 5% solvent B to 100% solvent B. The sample loading volume and flow were 20 μL and 8 μL/min, respectively. For protein fractions, a gradient of 60 min was employed, while for the initial crude fractions, a 90 min gradient was employed to increase analytical depth. For both methods, the remaining settings were identical. The analysis was run in a full MS/ddMS2 data-dependent mode with an MS1 scan range of 300–1600 m/z, positive polarity, and a default charge of 2. The MS1 resolution was 60,000, while the dd-MS2 resolution was 15,000. TopN, AGC target, and Maximum IT were set to 20, 1e5, and 45 ms, respectively. The isolation window, (N)CE, and dynamic exclusion window were set to 1.2 m/z, 28 eV, and 20.0 s, respectively. Peptide match was preferred, and isotopes were excluded.

#### 2.7.4. LC-MS/MS Data Processing

LC-MS/MS data were processed with MaxQuant v.2.2.0.0 [46]. The database for MaxQuant built-in Andromeda search engine [47] was built by selecting all proteins in the UniProt database from both the target species *L. perenne* (taxid = 4522) with 824 protein sequences (downloaded May 2nd 2024) and the reference proteome of the model species *Brachypodium distachyon* (UP000008810) with 45301 protein sequences (downloaded 2 May 2024). The processing with MaxQuant was performed with the oxidation of methionine and protein N-terminal methylation as a variable modification and cysteine carbamidomethylation as a fixed modification, while trypsin (up to 2 missed cleavages was allowed) was assigned as the specific protease. The allowed m/z deviation was 4.5 ppm. The minimum peptide size was set to be 7 amino acids and the maximum peptide size was set to 4600 Da. A false discovery rate of 1% was employed on both the peptide and protein levels. Matching between runs as well as dependent peptides was enabled. Quantification was performed by intensity-based absolute quantification (iBAQ) for all data and MaxLFQ (triplicate analysis) for the crude fraction data.

#### 2.7.5. Downstream Data Analysis of MaxQuant Data from SEC Fractions

Downstream processing of MaxQuant data for SEC fractions was performed with R version 4.4.1 (https://www.R-project.org/) with the following packages: tidyverse 2.0.0; dplyr 1.1.4; readxl 1.4.3; heatmaply 1.5.0; ggplot2 3.5.1; patchwork 1.2.0; and UniprotR_2.4.0.

The identified proteins were filtered for the removal of contaminants and reverse proteins. The protein data were then split into three groups containing lead proteins without fragments, lead proteins with fragment annotation but with other proteins of the same name within the group, or lead proteins with fragment annotation and without any full-length proteins in the group. For the latter two groups with lead proteins as fragments, the iBAQ was recalculated from the raw MS1 intensity and marked with a “*” in the protein name.

For lead protein fragments where there were lead proteins of the same name without fragment annotation the following formula was used:$${iBAQ}_{rec}=\frac{\sum I_{p}}{TP_{max}}$$
where *iBAQ_rec_* is the recalculated iBAQ for the protein group, *ΣI_p_* is the sum of peptide intensities for the protein group, and *TP_max_* is the maximum of theoretical peptides for a protein of the same name without fragment annotation within the protein group.

In the latter case, where the lead protein was annotated as a fragment and there were no full-length proteins of the same name within the group, a different approach was employed to correct iBAQ intensities. The theoretical number of peptides (*TP_max_*) was determined through either another lead protein with the same name in the dataset (but not in the protein group) or by identified proteins from a BLAST search on the fragment lead protein. The selected proteins from the BLAST search were aligned with the fragment’s amino acid sequence and the one with the best coverage and similarity was used for calculating the theoretical number of peptides and a recalculated iBAQ value (see Appendix A). The molecular weight of these proteins was changed to BLAST hit proteins, since this number corresponded to the number of theoretical peptides. All iBAQ recalculations were performed protein- and sample-wise.

#### Isoform Combination

Different UniProt IDs may be annotated with identical protein names and thus represent isoforms of the same protein with similar bioactivity/function. As a result, they receive individual iBAQ values, which may underestimate the abundance of a particular protein type/group and their overall significance when evaluating abundance distribution due to additive effects. To account for this, proteoforms with precisely the same names were merged under one name and their riBAQ values were summarized. The isomers were named after the protein annotation, which appeared first in alphabetical order, and the number of isoforms was written in parentheses.

#### Gene Ontology Analysis and Data Filtering

Gene Ontology (GO) annotations were imported using UniProt IDs with the UniProtR package. A list of all GO annotations was examined to pick out those relevant to antioxidant activity. These were used to filter the dataset for in silico antioxidant prediction. The annotations used were GO:0034599 (cellular response to oxidative stress), GO:0009055 (electron carrier activity), GO:0042542 (response to hydrogen peroxide), and GO: 0006979 (response to oxidative stress) as well as the keywords, “Heme binding”, “oxidoreductase activity, “glutathione transferase activity”, “electron transfer activity”, “peroxidase activity”, “lactoperoxidase activity”, ”metal ion binding”, “chaperone activity”, “iron”, “oxidative”, “superoxide”, “transition”, “proton”, “ferri”, and “ferre”.

The resulting subset was subsequently filtered in a three-step process. Firstly, the GO-positive hits were cross-referenced with the analysis on the initial crude fractions, filtering out the proteins that were not identified. Next, a confidence filtration step was applied to ensure that only high-quality identifications were considered. With reference to the triplicate analysis of the green juice (as this stream is relevant for the potential downstream development of a food-grade protein product), proteins were only considered if at least three of the following four criteria were met: (1) a protein-level Andromeda score of at least 40; (2) identified by MS/MS in at least two of three replicates (i.e., not just by matching); (3) a sequence coverage of at least 5% in at least two of three replicates; and (4) at least two unique peptides in at least two of three replicates. The final filtering step related to abundance in the fractionated green juice was based on the assumption that effective antioxidants must constitute a substantial amount of the protein to be considered relevant. As such, only proteins constituting at least 0.5% of the protein across all fractions (by TriBAQ) or at least 1% in any fractions were considered. The filtered subset was ultimately used as the basis for a thorough literature search to identify any reported information on their relation to antioxidant activity either *in vivo* or *in vitro*.

#### 2.7.6. Downstream Data Analysis of Crude Fractions

The identified proteins from the MaxQuant search were initially filtered for the removal of contaminants and false positives. Next, the data were inspected for reproducibility using an inclusion criterion of positive identification in at least two of three replicates for each sample. Venn diagrams were created by Venny v 2.1 (https://bioinfogp.cnb.csic.es/tools/venny/index.html (accessed on 5 November 2024)). Relative molar abundance was estimated by riBAQ quantification as described above.

MaxQuant LFQ data for triplicates of crude fractions were analyzed using Mass Dynamics 2.0 [48]. Missing values were imputed by NMAR with a mean position factor of 1.8 and an SD factor of 0.3. For pair-wise analysis, proteins were considered significantly and differentially abundant if the adjusted *p*-value was below 0.05 (i.e., false discovery rate (FDR) < 5%) and the fold change ratio was greater than 2 (i.e., log2 (FC) > 1).

### 2.8. Statistical Analysis

Statistical analysis was performed in GraphPad Prism (10.0.2) using ANOVA with a 95% confidence level. For dry matter and crude protein, ordinary ANOVA and multiple comparisons by Tukey were applied. For riBAQ protein abundance, Welch ANOVA (equal SDs not assumed) and multiple comparisons by Dunnett T3 were applied. For the enrichment analysis of MaxQuant LFQ data from the initial crude fractions, statistical analysis was performed in Mass Dynamics (2.0) as described above. For hierarchical clustering and heatmap representation, a Euclidean distance of 3 and row-wise Z-score normalization were applied for proteins identified as differential by the built-in ANOVA analysis.

## 3. Results and Discussion

### 3.1. Protein Characterization of Wet Fractionation

Since the method of primary processing influences mass balances, the three fractions, grass, pulp, and juice, were analyzed in relation to DM and CP (Table 1).

The absolute mass distributions after wet fractionation for pulp and juice were similar; however, approximately 10% of the absolute mass (7.9 g) was lost. This was ascribed to predominantly the loss of water as the DM and CP balance was practically void of any loss (0.1% and 0.3%, respectively). Approximately 41% of the CP and 37% of the DM ended up in the juice. As a result of wet fractionation, the pulp became significantly enriched (*p* < 0.03) in DM, while the juice was significantly enriched for CP compared to unprocessed grass (*p* = 0.030) and pulp (*p* = 0.003). The CP and DM distributions were similar to the literature, where the shares of CP and DM for green juice were reported as 40–60% and 30–50%, respectively [13]. Additionally, the CP% was less compared to the literature, where the CP for ryegrass, pulp, and juice was 16.7 ± 2.7%, 16.4 ± 3.5%, and 15.1 ± 3.9%, respectively [7]. These variations in the literature also indicate differences between harvests, which could influence protein profiles.

The protein profiles of grass, pulp, and juice were analyzed by SDS-PAGE under both reduced and non-reduced conditions to investigate differences in migration patterns (Figure 1A). Across all samples and conditions, a complex distribution and composition of proteins were detected. Notably, bands at approximately 50 kDa and 15 kDa were seen at a much higher intensity in the raw biomass compared to pulp and green juice. These correspond well with the expected monomeric bands of RuBisCO (hexadecameric complex: 560 kDa; large subunit: 56 kDa; small subunit: 16 kDa [18]). This indicated that the primary processing may induce the aggregation or increased formation of the hexadecameric complex. This was further substantiated by the reduced band intensities under non-reducing conditions where a higher proportion of protein was also observed at the top of the wells, as the size of the complex was above the resolving range of the gel. While the focus of grass protein research so far mainly has been on RuBisCO, these protein profiles also showed the intrinsic complexity of the ryegrass proteome and the presence of other proteins, which could have nutritional or functional value.

LC-MS/MS analysis provided more detailed profiles of the grass, pulp, and juice fractions in terms of identification and proteins not detectable by SDS-PAGE. After the filtering of common contaminants (9) and false positives (14), a total of 1084 protein groups were identified across all samples at 1% FDR (Appendix B). Hereof, 860, 885, and 875 protein groups were quantified in at least two of three replicates in grass, pulp, and juice, respectively. Among these, the majority (762 protein groups) were identified in at least two replicates in all three fractions (Figure 1B). The slightly higher number of protein IDs within the pulp and juice fractions may be a result of the mechanical processing during wet fractionation, providing increased accessibility of certain proteins. Not surprisingly, the majority of the identified proteins consisted of RuBisCO, which accounted for 34.6–38.2%, 32.9–35.0%, and 34.9–39.3% of the molar abundance (by riBAQ) in grass, pulp, and juice, respectively, when combining all isoforms of its small and large subunits (Appendix B). As the cumulative RuBisCO abundance was comparable across the crude fractions, this further suggests significant changes in the complex and/or aggregation state when comparing monomeric band intensities from SDS-PAGE analysis (Figure 1A). In the crude fractions, various photosystem proteins (12.2–15.1%), ATP synthase subunits (5.6–7.4%), and chlorophyll a-b binding proteins (4.6–9.2%) were also identified as major constituents (Appendix B).

While the protein identifications from LC-MS/MS analysis showed a high level of comparability between the fractions (Figure 1B), in line with SDS-PAGE analysis (Figure 1A), differences were also apparent. This was further substantiated through, e.g., PCA analysis, where the three crude fractions cluster explicitly (Appendix A). To investigate this further and in a quantitative manner, an enrichment analysis was performed using normalized LFQ intensities (Figure 2). When comparing the juice to the raw biomass, five proteins were significantly enriched (log2 fold change > 1; 5% FDR threshold) while four were depleted (Figure 2A). Comparing the pulp to the raw biomass (grass), seven proteins were significantly enriched while one was significantly depleted (Appendix A). In the last pair-wise comparison of juice to pulp, four proteins were significantly enriched while 13 were significantly depleted (Appendix A). Of these 34 differentially abundant proteins between crude fractions (Appendix A), it should be noted that the majority (23) relied on imputed values, which is why the fold change should be interpreted with some caution. Notably, four of the seven proteins enriched in the pulp compared to the raw grass (two beta-glucosidases (I1GNA1 and I1GNA2), Cytochrome c domain-containing protein (I1I9E4), and Pectinesterase (I1HEX8)) were also found to be enriched in the pulp compared to the juice. This indicated a degree of selectivity during wet fractionation, with these proteins preferentially localized in the residual pulp. This observation was further supported by the differential heatmap representation (Figure 2B), which revealed four distinct clusters with specific enrichment patterns across the fractions. The heatmap included 80 protein groups identified as differentially abundant by the ANOVA analysis of normalized LFQ intensities, providing a comprehensive overview of differential protein distribution across all three fractions. For the remaining 1004 protein groups, no statistically significant abundances were observed across the three crude fractions (Appendix A). Moreover, replicates of fractions were grouped by hierarchical clustering, which was further validated through a PCA analysis of data variance (Appendix A). These results indicate a high degree of reproducibility in upstream sample preparation using the iST kit for plant tissue.

### 3.2. Protein Fractionation and Selection

The filtered green juice containing soluble protein was separated by size exclusion chromatography, which yielded 62 fractions (Figure 3A). From the calibration experiment, the void volume was determined to be 94.7 mL (Appendix A). A massive peak in 280 nm UV absorption was detected in fractions 4–8. While the apex could not be identified, and due to partial overlap with the void volume, an MW estimation could not be explicitly performed, the MW range nevertheless corresponded well with the presence of a multimeric form of RuBisCO of 560 kDa in the green juice, corroborating the indications hereof from SDS-PAGE and initial quantitative proteomic analysis. The presence of multimeric RuBisCO after pressing green biomass prior to any downstream processing has recently been reported from spinach [27] and alfalfa [49]. Two other peaks in 280 nm UV absorption were detected towards the end of the chromatogram, which are expected to be small aromatic compounds, e.g., polyphenols, due to the high 260/280 ratio and the low estimated MW based on column calibration.

All 62 fractions were screened for DPPH radical scavenging activity, which served as the basis for selecting 15 fractions for further investigations of antioxidant activity (Figure 3B). The first six and last 30 fractions exhibited evident activity under equivolumetric conditions (Appendix A), which may reflect the high content of proteins and small aromatic compounds, respectively, within these fractions based on high A280 (Figure 3A). However, the fractions were selected to cover all parts of the chromatogram, since the protein concentrations were not standardized in the crude screening assay. Late eluting fractions were included despite the presumed composition of small aromatic compounds. Iron chelation screening could not be performed due to the chelation of ferrous iron by the phosphate buffer.

### 3.3. Ex Vivo Antioxidant Activity

Two *in vitro* assays, DPPH radical scavenging and iron chelation, were conducted on the selected fractions to evaluate the *ex vivo* antioxidant capacity of the green juice. The activity was evaluated based on the EC50 value, which is defined as the concentration of antioxidants that caused a 50% decrease in absorbance. Thus, the lower the EC50 value, the higher the DPPH radical scavenging or iron-chelating ability of the antioxidant. Ideally, the experimental data to calculate an EC50 value should fit a sigmoidal curve [50]. Due to narrow concentration spans, only parts of the sigmoidal curve were observed, and therefore a linear fit was used for DPPH radical scavenging and a logarithmic fit for iron chelation (Appendix C). Additionally, certain protein fractions were excluded due to the unavailability of curve fitting, despite the observation of activity.

Since the assays were normalized based on protein mass concentration, the interpretation of activity could be erroneous due to differences in protein size, which conflicted with the stoichiometry in the reaction. Hence, EC50 values (Table 2), were converted to molar concentrations by using experimental calibration and estimated MW for each fraction to assess activity based on molar concentration rather than mass concentration. Due to the limited availability of lyophilized fractions, replicates could not be performed. As the positive controls exhibited variation, identifying the most active protein fractions was associated with some uncertainty. It should be noted that antioxidant activity was quantifiably measured among closely adjacent fractions, which were assumed to have similar molecular composition. These fractions displayed comparable EC50 values and overall trends were consistent. This indicates the presence of antioxidant proteins with the potential to prolong the oxidation period of the green juice. Therefore, determining the quantitative protein-level composition in each fraction was of particular interest.

### 3.4. Overview of Protein Composition and Most Abundant Proteins

The protein composition of the selected protein fractions was analyzed by LC-MS/MS. From the MaxQuant database search, 1277 protein groups matched the UniProt entries for *L. perenne* on the taxonomic level and the reference proteome of *Brachypodium distachyon*. After accounting for protein fragments as well as filtering contaminants and false positive protein IDs, 1204 proteins remained (Figure 4, Appendix D). iBAQ values for each protein group were calculated by MaxQuant and are a measure of protein abundance, where raw intensities were divided by the number of theoretically observable peptides. Thus, iBAQ values are proportional to the molar quantities of the proteins [51,52]. As the iBAQ values were not normalized, the abundance of proteins cannot be compared across fractions. Only the presence of proteins and how much of the total protein within a fraction they constituted can be compared.

A diverse range of proteins was identified in each fraction, with a general trend indicating that earlier fractions contained a greater diversity of proteins compared to later fractions (Figure 4). Fractions that eluted closely together displayed similar protein profiles, forming distinct clusters. For example, fractions 3, 5, and 7 showed higher similarity to one another than to later fractions. This corroborated similar EC50 values for adjacent SEC fractions (Table 2). Despite the detected clustering, certain proteins were detected across multiple fractions, with some even appearing in all fractions, suggesting inefficient separation during fractionation. While certain proteins were consistently detected across fractions, each fraction maintained a unique protein composition.

To investigate the most abundant proteins across the juice fractions, proteins with a TriBAQ (i.e., total riBAQ) value above 0.5% were identified (Figure 5). Proteins presented as isomers were characterized as distinct entities in UniProt but were manually combined if they shared identical protein names, giving 859 unique protein annotations from the 1204 proteins (Appendix E). This combination aimed to give a more realistic view of abundance for specific protein classes based on assumed similarity in functionality.

Across all investigated fractions, 36 proteins (by annotated name) were found above the 0.5% TriBAQ threshold. The most abundant proteins were the large and small subunits of RuBisCO. This is consistent with the quantitative analysis of the unfractionated juice (Appendix B) and the fact that RuBisCO is the most abundant leaf protein across plant species. In general, most of the 36 abundant proteins were associated with photosynthesis, cellular respiration, or other fundamental cellular processes. Additionally, proteins involved in oxidative stress response, such as thioredoxin-dependent peroxiredoxin (TPx), superoxide dismutase (SOD), glutaredoxin-dependent peroxiredoxin (GPx), and different peroxidases (Px), were identified as highly abundant.

The riBAQ values of the 36 proteins in each fraction were analyzed to examine the distribution of the most abundant proteins across the fractions. Certain proteins were detected in multiple fractions. For instance, the RuBisCO large subunit was found in all fractions with a high abundance in both the first and the last fractions. It was expected that RuBisCO subunits would appear in high abundance in the initial fractions, due to the size of the native multimeric complex; hence, their presence in the later fractions was unexpected. However, it is important to recognize that the abundance values reflected the riBAQ measurements, meaning that the specific quantities may vary even if their relative shares remain consistent. Examination of the sequence coverage and SDS-PAGE of the 15 selected protein fractions (Appendix A) showed that, although the RuBisCO large subunit was found by riBAQ to be abundant in the later fractions, the sequence coverage was low, and a no distinct bands corresponding to the RuBisCO large subunit was detected by SDS-PAGE analysis of the last fractions. Thus, peptides stemming from the endogenous proteolysis of the RuBisCO large subunit might have been inaccurately presented as the whole protein. Moreover, the expected MW for the later fractions makes it unfeasible for the RuBisCO large subunit to be located here. This underscores the importance of careful data interpretation and using different protein characterization methods to prevent the misinterpretation of the protein composition of the fractions.

Interestingly, a high abundance of chitinase was also detected in the later eluting fractions (Figure 5). The SDS-PAGE analysis of the protein fractions revealed distinct bands around 30 kDa in fractions 43–55 (Appendix A), which was unexpected, given that the MW in these later fractions was anticipated to be below 1 kDa (Table 2, Figure 5A). Chitinase from the model organism *B. distachyon* has an MW of approximately 34 kDa, including a 22-residue signal peptide, (UniProt AC# I1HQL2), which corresponds with the band observed by SDS-PAGE. Chitinase, which catalyzes the breakdown of the β-1,4-glycoside bond of N-acetyl-D-glucosamine in chitin [53], could have interacted with the dextran–acrylamide matrix of the SEC column. Although not entirely identical, it resembles the polysaccharide structure of chitin, thus potentially facilitating this interaction.

As a high global abundance (TriBAQ) threshold (Figure 5) may introduce bias and filter out proteins of substantial abundance within just one or a few fractions, an alternative method was employed to identify potentially antioxidant proteins. Rather than focusing on the most abundant proteins across all fractions, fraction-wise relative quantification (by riBAQ) was performed under the assumption that the most abundant proteins were responsible for or contributed to the measured antioxidant activity. Employing a threshold of riBAQ > 2% in any fraction, 43 protein classes (by name) were identified (Figure 6).

Of the 36 proteins identified using global abundance (Figure 5), 34 were also identified using fractional abundance (Figure 6), amounting to a total list of 45 proteins. As such, fractional analysis resulted in an additional 11 abundant proteins. Only two ribosomal proteins (A0A0Q3M5D8 and I1HQ35) were not above the fractional threshold. Of the additional 11 proteins identified by a fractional abundance threshold, the majority were affiliated with carbohydrate hydrolysis, and none appeared to have an affiliated activity that could relate to antioxidative properties. Ultimately, the fractional abundance filtering did not provide additional insights into the potential effectors of the observed *in vitro* activity.

### 3.5. Prediction of Antioxidant Proteins Using GO-Term Analysis

Identified proteins from the fractional analysis were also investigated by GO-term analysis to predict potentially responsible antioxidant proteins. The 859 proteins were filtered based on a subset of molecular function GO annotations yielding 166 unique proteins by name (representing 235 lead protein accessions) with a diverse range of molecular functions (Appendix A). This list of lead proteins was then subjected to multi-level filtering. Initially, GO-predicted lead proteins were cross-referenced with proteins identified in the initial crude fractions, filtering out those that were not identified in this analysis (83 lead proteins). Next, we applied a confidence filter to increase the certainty of identification, filtering out 59 low-confidence identifications. Retaining the assumption that the proteins responsible for the observed activity must be abundant, we applied an abundance filtration. However, this filtration combined both previous approaches while decreasing abundance thresholds (riBAQ > 1% in any fraction or TriBAQ > 0.5%). This left a final shortlist of 18 lead proteins, representing 14 unique proteins (Table 3). We next performed a thorough literature investigation to identify proteins with explicitly reported antioxidant activity and used this to categorize the proteins based on direct (A), indirect (I), and no explicit (N) antioxidative function (Table 3).

Four proteins (ATP synthase, ATPase, and two Calvin cycle enzymes) were classified as non-antioxidants (N), while another four proteins (ferredoxin-NADP reductase (FNR), lactoylglutathione lyase (Glyoxalase I; Glo1), glutathione S-transferase (GST), and a putative dehydroascorbate reductase (DHAR)) are classified as indirect antioxidants (I), as they are involved in the formation or regeneration of small-molecule antioxidants *in vitro* and thereby require the presence of these to facilitate antioxidant capacity. It should be noted that since the UniProt protein database was extracted, the putative DHAR is now annotated as a GST isoform. However, they fall within the same category. The six remaining proteins were found in the literature to have direct antioxidant activity (A). These were L-ascorbate peroxidase (APX), glutaredoxin-dependent peroxiredoxin (GPx), thioredoxin-dependent peroxiredoxin (TPx), superoxide dismutase (SOD), peroxidase (Px), and the uncharacterized protein ycf33, which showed similarity to peroxiredoxin Q (Table 3). Apart from APX, DHAR, and ycf33, all direct and indirect antioxidant proteins were also identified through the abundance-based filtration approach, through which no other proteins of immediate interest were found. Consequently, the subsequent analysis focused on the ten proteins within these two categories.

### 3.6. Correlating Protein Abundance and GO-Term Analysis with In Vitro Antioxidant Activity

Fractions 5 and 7 exhibited the lowest molar EC50 values in the DPPH RSA assay, indicating the highest antioxidant activity. Both fractions showed a high abundance of RuBisCO- and ATP synthase-related proteins, as previously described. A similar protein profile was seen in fraction 3, though its molar EC50 value was not calculated due to the lack of an estimated size (eluting in the void volume). However, based on mass EC50 value, fraction 3 aligned closely with fractions 5 and 7, suggesting a comparable antioxidant effect across the early fractions. RuBisCO and ATP synthase proteins, in their native forms, are generally not described as antioxidants in the literature. Nevertheless, known antioxidant proteins such as FNR and TPx were also identified in these fractions (particularly in fractions 5 and 7), with notable abundance (0.29–0.47% and 0.61–1.18%, respectively). In fraction 3, most proteins of interest were identified as part of the column flow-through, which is why the observed activity in this fraction likely cannot be ascribed to specific proteins. While the measured *in vitro* activity may originate from these proteins, further purification and isolation would be required to validate their antioxidant potential *ex vivo.* While RSA activity was also detected in late eluting fractions and lower EC50 values (by mass concentration) were determined, this activity was ascribed to smaller molecules or metabolites from *L. perenne* based on the estimated MW within these fractions. In this respect, it should be noted that the MW estimates fell outside the resolving range of the column, leading to significant uncertainty in the direct MW estimation.

In the iron chelation assay, fractions 7–22 exhibited the lowest molar EC50 values, indicating their high chelating efficiency. However, despite unavailable curve fitting, protein fractions 3 and 5 should not be disregarded, as they also demonstrated notable activity in the assay (Appendix C). Consequently, when analyzing patterns in protein abundance, these fractions should be included in the assessment. As RuBisCO is reported to chelate divalent Mg within the active site as an essential component for the enolase activity [69], this may also facilitate the chelation of divalent iron as measured in the assay. As such, RuBisCO itself may display intrinsic chelating activity and could add to the oxidative stability of a product. Moreover, a large proportion of the predicted antioxidant proteins are also localized in this range of fractions. For instance, GST was enriched in fraction 19 while predicted antioxidant proteins such as TPx, SOD, APX, and FNR were highly enriched in fraction 22, thereby indicating their potential role in iron scavenging, eliminating its pro-oxidative effect. In contrast, GPx, DHAR, ycf33, and Px proteoforms were enriched in later eluting fractions (28–34), where lower activity was measured. This indicated that these proteins were likely less responsible for iron scavenging than other antioxidant proteins predicted through GO annotation. Although several antioxidant proteins were present in the active fractions, there was no clear correlation between activity and specific protein abundance across the fractions. This suggests that the relationship between protein content and antioxidant activity may be rather complex and cannot be ascribed to single proteins but rather cumulative and/or agonistic effects.

Both approaches have inherent limitations, making their combination an ideal approach. In silico prediction is subject to potential bias and the inconsistency of which relevant properties were included in the GO annotations for the specific proteins resulted in uncertainty in estimating antioxidant capacity. An example of this is FNR, which uses iron as a co-factor but was not annotated as ion-binding [70]. Additionally, focusing solely on the most abundant protein in the highly active protein fraction potentially could dismiss proteins that may still have antioxidant effects even if their relative abundance falls below the applied threshold. Furthermore, only testing electron and hydrogen transfer and iron chelation mechanisms excludes other mechanisms that might contribute to the antioxidant capacity in the green juice.

Nonetheless, the combination of the two antioxidant identification methods resulted in ten different known proteins related to antioxidant activity. However, whether these proteins were responsible for the measured *ex vivo* activity is somewhat ambiguous. The prediction methods were based on the literature and annotations of cellular functions *in vivo*, which is why similar activity may not be retained *ex vivo*. As the proteins were not in their original cellular compartment, the surrounding environment, as well as other proteins and co-factors, showed diminished activity despite being recognized as antioxidants in the literature. In contrast, proteins with no known antioxidant activity *in vivo* may exert *in vitro* activity when isolated. To investigate this further, a more stringent isolation and purification of individual proteins would be required, allowing the evaluation of their isolated activity *ex vivo*.

### 3.7. Abundance of Known Antioxidant Proteins in Crude Fractions

While the identification of the proteins responsible for the measured antioxidant activity was associated with uncertainty, our analysis revealed substantial abundances of strong candidate proteins in the fractions showing *in vitro* antioxidant activity. Consequently, we investigated the dataset from the three initial crude fractions to evaluate the overall abundance and potential of the ten identified direct and indirect antioxidants (Figure 7A).

The ten protein classes were found in the unprocessed biomass with an average total abundance of 1.63% by riBAQ. Of the ten classes, FNR was the most abundant (0.47%), followed by TPx (0.34%). Except for the combined proteoforms of APX (0.14%), SOD (0.13%), and GPx (0.12%), the remaining groups were found at an average of 0.10% or lower. Comparing abundances across the initial crude fractions (Figure 7B), TPx, SOD, FNR, and GPx were significantly (*p* < 0.05) enriched in the juice compared to both raw biomass and pulp. Glo1 was significantly (*p* = 0.017) depleted in the pulp while no significant difference was found between raw biomass and green juice. Ycf33 and GST were significantly (*p* = 0.010 and *p* = 0.040, respectively) enriched in the juice compared to the pulp, while no significant differences were detected in comparison to the raw biomass. No significant differences were observed for APX, Px, and DHAR across crude fractions. Cumulatively, a significant (*p* = 0.0021) increase of 23.9% was detected in the juice, while a smaller and not significant decrease of 2.8% was determined in the pulp when compared to the raw biomass. Taken together, these findings showed that the identified antioxidant protein candidates were enriched in juice following wet fractionation and that this was especially attributed to higher accumulation and partitioning in the liquid phase of TPx, SOD, FNR, and GPx. Based on the *in vitro* assays, three of these proteins (TPx, SOD, and FNR) were enriched in fractions with high chelating activity. As they were furthermore the four most abundant candidate proteins (constituting ~1.5% of the molar protein abundance in the green juice), it is likely that these classes were responsible for the measured antioxidant activity of the fractionated green juice *ex vivo* and that strategies for their enrichment within a process stream may improve the oxidative stability and ultimately the shelf life of the end product.

## 4. Conclusions

The green biorefining of leafy greens such as perennial ryegrass (*L. perenne*) is currently undergoing major advancements for transforming the inedible biomass into a protein-rich ingredient suitable for human consumption. One of the major challenges remaining is imposed by the dynamic nature of the green juice after pressing, where proteins, enzymes, and phytochemicals are decompartmentalized, leading to a highly reactive environment. To alleviate this, a basic understanding of the endogenous antioxidative capacity of the biomass is pivotal. Here, we employed size exclusion chromatography for the fractionation of the grass juice to gain a deeper understanding of the *ex vivo* antioxidant properties. Through the *in vitro* assaying of *L. perenne* green juice fractions for radical scavenging and iron chelation, we linked the measured *in vitro* activity to the major protein constituents using quantitative LC-MS/MS-based proteomics. Through GO-term analysis, we identified at least ten enzymes known to be involved in *in vivo* antioxidant processes, indicating that their activity was maintained *ex vivo*. Moreover, we showed that their cumulative abundance was significantly enriched by 24% in the green juice (2.0% riBAQ) compared to the unprocessed ryegrass (1.6% riBAQ), while they were slightly depleted in the resulting pulp. Moreover, the measured activity coincided with an enrichment of, especially, TPx, SOD, and FNR in fractions with highly detected ferrous chelation. However, their isolated activity *in vitro* still requires validation to verify the retention of antioxidative properties *ex vivo* following wet fractionation.

This study also provides the, to date, most detailed characterization of the *L. perenne* proteome and quantitative protein partitioning in the juice and pulp fractions after wet fractionation. Our analysis further revealed that the major protein of the biomass, RuBisCO, primarily exists in the native heterohexadecameric form after pressing. This implies that it may be effectively separated from major drivers of enzymatically driven oxidation processes and subsequent phytochemical protein modification and cross-linking using size-based fractionation in a biorefinery concept, ultimately reducing unwanted side reactions in the juice. These findings open possibilities for targeted process development and optimization to improve the oxidative stability of the green juice during green biorefining, leading to a higher-quality protein product.

## Figures and Tables

**Figure 1 proteomes-13-00008-f001:**
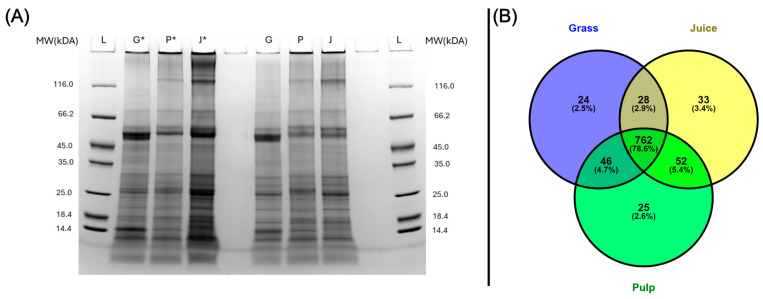
Protein distribution in grass (G), pulp (P), and juice (J). (**A**) SDS-PAGE analysis performed under reduced (*) and non-reduced conditions. (**B**) Venn diagram of protein groups quantified in at least two of three replicates for grass (blue), pulp (green), and juice (yellow) by LC-MS/MS analysis.

**Figure 2 proteomes-13-00008-f002:**
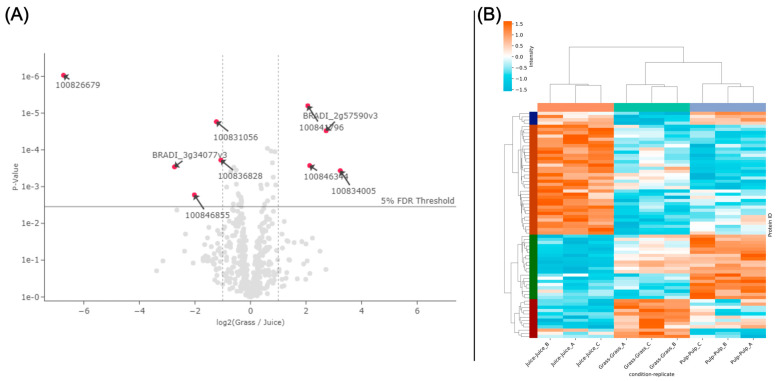
Differential analysis of proteins across the three crude fractions. (**A**) Volcano plot of differentially abundant proteins in a pair-wise comparison of grass and juice. Differentially abundant proteins (log2 fold change > 1, adjusted *p*-value < 0.05 (5% FDR threshold)) are indicated by arrows and red dots and annotated by gene name (see Appendix A). (**B**) Heatmap representation of differentially abundant proteins (rows) across replicates of all three crude fractions. In the heatmap, normalized LFQ intensities are standardized using row-wise Z-scores, and proteins are clustered by similarity using a cluster distance of three. Crude fractions and replicates (columns) are clustered hierarchically.

**Figure 3 proteomes-13-00008-f003:**
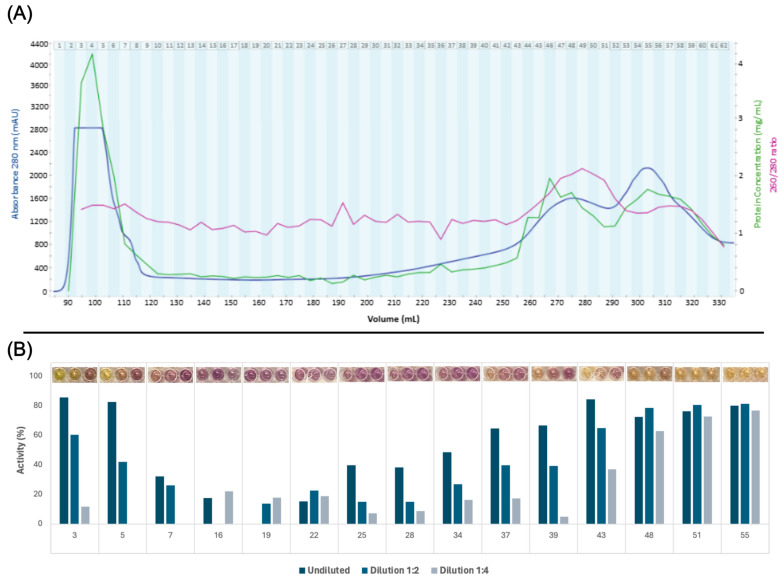
Production and selection of ryegrass green juice fractions. (**A**) Size exclusion chromatogram, protein concentrations, and 260/280 ratios. Size exclusion chromatogram showing absorbance at 280 nm relative to elution volume. The upper limit for absorbance measurements was 2800 mAU. Each collected fraction is denoted by number. Corresponding measurements of protein concentration (by A280, 1 A = 1 mg/mL) and A260/280 ratio obtained via UV spectroscopy for each fraction are also shown. (**B**) Activity of selected fractions for each dilution in the DPPH radical scavenging screening. Pictures of the assay results are seen above the graphs for each fraction. The background color of some protein fractions influenced absorption, and therefore certain data points are excluded due to negative activity values.

**Figure 4 proteomes-13-00008-f004:**
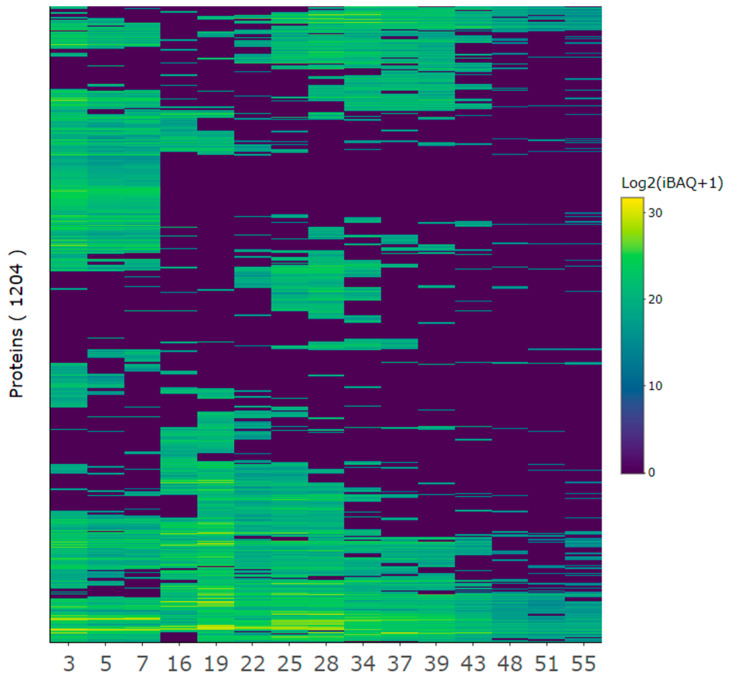
Quantitative overview of protein composition for selected protein fractions. The protein iBAQ (+1 to avoid infinity numbers) within each fraction was log2-transformed to better illustrate the broad dynamic range of protein abundances and represented in a heatmap from low (blue) to high (yellow) abundance to inspect protein-level composition and variability between SEC fractions.

**Figure 5 proteomes-13-00008-f005:**
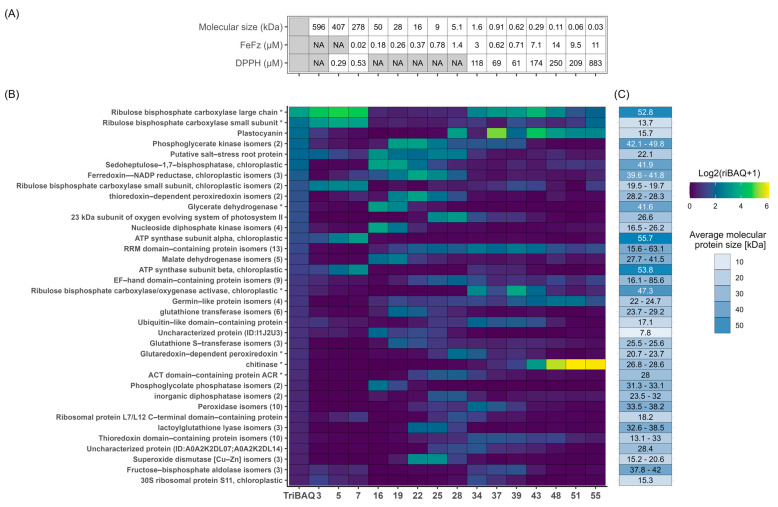
Fractional distribution of the most abundant proteins across all selected fractions. (**A**) Metadata for selected fractions, showing molecular size in kDa estimated from SEC and the EC50 values from the DPPH and FeFz assay. Shaded boxes represent no available size or EC50 due to either being a summary metric (TriBAQ) or that no value was obtained (“NA”, see Table 2). (**B**) Distribution and abundance of the most abundant proteins (TriBAQ > 0.5%) across selected fractions. The TriBAQ column corresponds to the total relative abundance of proteins, all fractions combined. The gradient for fraction columns corresponds to the riBAQ of filtered proteins within each fraction from low (blue) to high (yellow). The protein riBAQ within each fraction was log2-transformed and +1 was added to avoid infinity numbers. The UniProt AC# of uncharacterized proteins and the numbers of protein isomers are presented in parentheses. Proteins with an (*) indicate that the lead protein was a fragment and the iBAQ was therefore recalculated. (**C**) The blue gradient shows a broad MW range, while the number in each cell shows the actual MW or the MW range of grouped isoforms. For RuBisCO small subunits and glycerate dehydrogenase, the MW of the best BLAST hit is used, since this size was used for the recalculation of their iBAQ and riBAQ values.

**Figure 6 proteomes-13-00008-f006:**
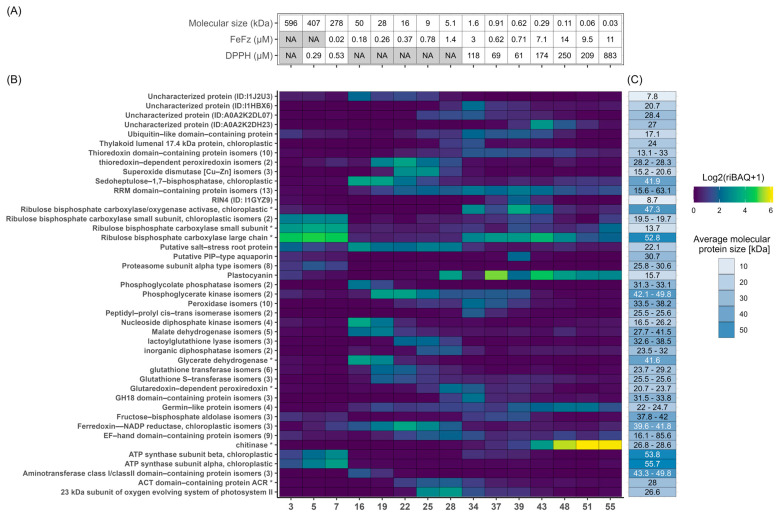
Fractional distribution of the most abundant proteins across each selected fraction. (**A**) Metadata for selected fractions, showing molecular size in kDa estimated from SEC and the EC50 values from the DPPH and FeFz assay. Shaded boxes represent no available EC50 as no value was obtained from curve fitting (“NA”, see Table 2). (**B**) Relative quantification and fractional distributions of proteins with riBAQ > 2.0% within at least one of the 15 selected fractions from low (blue) to high (yellow) abundance. Proteins with an (*) indicate that the lead protein was a fragment and the iBAQ was therefore recalculated. (**C**) Protein MW of the identified proteins, with RuBisCO small subunit and glycerate dehydrogenase MW corresponding to their BLAST hit, since these were used for the recalculation of iBAQ and riBAQ values.

**Figure 7 proteomes-13-00008-f007:**
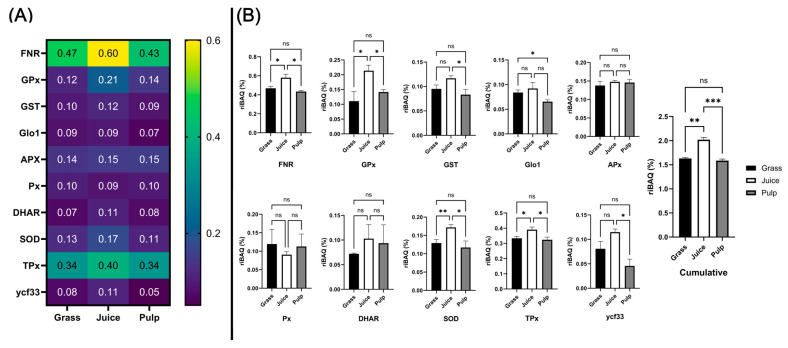
Enrichment analysis of candidate antioxidant proteins across initial crude fractions. (**A**) Heatmap (blue to yellow) showing average (n = 3) relative molar abundance (by riBAQ) of the eight candidate protein families (identified by GO-term analysis) in unprocessed ryegrass, green juice, and residual pulp. (**B**) Histograms and statistical analysis of riBAQ abundance of the eight candidate families and the cumulative abundance of all candidate proteins across unprocessed ryegrass (black), green juice (white), and residual pulp (grey). Abundance is indicated as the mean with the standard deviation (n = 3). Statistical analysis is performed by one-way ANOVA (Welch) and Dunnett T3 correction and the significance level (from adjusted *p*-values) is indicated by “ns” (*p* > 0.05), “*” (*p* ≤ 0.05), “**” (*p* ≤ 0.01), and “***” (*p* ≤ 0.001).

**Table 1 proteomes-13-00008-t001:** Mass balance and crude protein content of *L. perenne* grass, pulp, and green juice. Values are given in average ± standard deviation (n ≥ 3).

Crude Fraction	Mass (g)	DM (% *w*/*w*)	DM Distribution	Crude Protein ^1^ (%, DM Basis)	Crude Protein Distribution
Grass	75.2	19.2 ± 1.1 ^a^	100%	11.2 ± 0.4 ^a^	100%
Pulp	32.9	27.7 ± 3.1 ^b^	63.0%	10.4 ± 0.4 ^a^	58.7%
Green Juice	34.4	15.5 ± 2.1 ^a^	36.9%	12.5 ± 0.5 ^b^	41.0%

^1^ Crude protein is based on crude nitrogen using an N-to-protein conversion factor of 6.25. Superscript letters refer to significant (*p* < 0.05) differences between the means of each column by ANOVA and multiple comparisons by Tukey.

**Table 2 proteomes-13-00008-t002:** EC50 values of DPPH RSA and ICA and estimated average MW for the selected SEC fractions. Trolox and EDTA (positive controls) EC50 is given in average ± standard deviation (n = 3).

		DPPH		Iron Chelation	
Protein Fraction	MW_avg_ (kDa)	EC50 (μg/mL)	EC50 (μM)	EC50 (μg/mL)	EC50 (μM)
3	n.d. ^1^	130	n.d. ^1^	n.d. ^2^	n.d. ^1,2^
5	407	120	0.29	n.d. ^2^	n.d. ^2^
7	278	150	0.53	6.5	0.023
16	50.0	n.d. ^2^	n.d. ^2^	9.0	0.18
19	28.2	n.d. ^2^	n.d. ^2^	7.4	0.26
22	15.9	n.d. ^2^	n.d. ^2^	5.9	0.37
25	9.00	n.d. ^2^	n.d. ^2^	7.0	0.78
28	5.08	n.d. ^2^	n.d. ^2^	7.2	1.4
34	1.62	191	120	4.8	3.0
37	0.914	63	69	0.57	0.62
39	0.624	38	61	0.44	0.71
43	0.291	51	170	2.1	7.1
48	0.112	28	250	1.6	14
51	0.0634	13	210	0.60	9.5
55	0.030	26	880	0.31	11
Trolox	0.251	37 ± 18	150 ± 74		
EDTA	0.292			2.5 ± 1.5	8.5 ± 5.1

^1^ Molar EC50 could not be determined (n.d.) as the fraction was part of the void volume. ^2^ EC50 could not be determined (n.d.) as the the obtained data point did not allow for satisfactory curve fitting.

**Table 3 proteomes-13-00008-t003:** Overview of the 14 unique proteins identified from GO-term analysis and subsequent confidence and abundance filtering. For each specific protein, the UniProt AC# of the lead protein from the protein group (from MaxQuant) is listed. Isoforms and subunits have been grouped. Proteins are categorized (Cat.) as being antioxidant (A), indirect antioxidants (I), or non-antioxidant (N) based on their mechanism of action as described in the literature.

Protein	Lead AC#s	Cat.	Mechanism	Ref.
ATP synthase (α, β, and γ subunits)	A8Y9G7 A8Y9H7 I1GU08	N	Formation of ATP	[54]
ATPase (V-type α subunit)	I1GVU2	N	Formation of ADP by ATP hydrolysis	[55]
Thioredoxin-dependent peroxiredoxin (TPx)	I1IXL1 I1IAA5	A	Reduction of hydrogen peroxide and hydroperoxides.	[56,57]
Superoxide dismutase (SOD)	I1I9J4	A	Oxygen radical scavenging	[58]
Ferredoxin-NADP reductase (FNR)	I1H1Z5 I1HW30	I	NADPH regeneration	[59]
Lactoylglutathione lyase (Glyoxalase I, Glo1)	A0A0Q3HLX8	I	Glutathione formation	[60]
L-ascorbate peroxidase (APX)	H6BDN2	A	Reduction of hydrogen peroxide	[61]
Glutaredoxin-dependent peroxiredoxin (GPx)	I1HY81	A	Reduction of hydrogen peroxide	[62]
Glutathione S-transferase (GST)	A0A165FYU2	I	Reduction of lipid hydroperoxides by facilitating GSH binding	[63]
Peroxidase (Px)	I1HZ46	A	Reduction of hydrogen peroxide	[64]
Putative dehydroascorbate reductase (DHAR)	H6BDN5	I	Regeneration of ascorbate	[65]
Peroxiredoxin Q-like (ycf33) ^1^	A0A0Q3H912	A	Reduction of alkyl hydroperoxides	[66]
Ribulose-phosphate 3-epimerase (RPE)	I1H9A1	N	Calvin cycle enzyme	[67]
Sedoheptulose-1,7-biphosphatase (SBP)	I1HTG2	N	Calvin cycle enzyme	[68]

^1^ Uncharacterized protein ycf33 shows high similarity to peroxiredoxin Q by BLAST analysis.

## Data Availability

The mass spectrometry proteomics data have been deposited into the ProteomeXchange Consortium via the PRIDE [71] partner repository with the dataset identifiers PXD058233 (crude fractions) and PXD058237 (SEC fractions), available at https://doi.org/10.6019/PXD058233 and https://doi.org/10.6019/PXD058237, respectively. Proteomics data output from MaxQuant and processed for filtering of contaminants and false positives as well as relative quantification by riBAQ can be found in the Appendix A for crude fractions (Appendix B), SEC fractions (Appendix D), and SEC fractions after isoform combination and riBAQ recalculations (Appendix E). Curve-fitting data for EC50 determinations are available in the Appendix A (Appendix C). All other data will be made available upon request.

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
