# Peer review of "Identifying Endogenous Proteins of Perennial Ryegrass (Lolium perenne) with Ex Vivo Antioxidant Activity"

_proteomes, 2025, doi:10.3390/proteomes13010008_

Round 1

Reviewer 1 Report

Comments and Suggestions for Authors

Conclusion:

In this study, Kathrine et al. employed size exclusion chromatography to fractionate the grass juice, aiming to gain a deeper understanding of its ex vivo antioxidant properties. They combined this with quantitative bottom-up proteomics, GO-term analysis, and fraction-based enrichment to provide the most detailed characterization of the L. perenne proteome and quantitative protein partitioning in the juice and pulp fractions. Their findings open possibilities for targeted process development and optimization to improve the oxidative stability of the green juice during green biorefining, leading to a higher-quality protein product.                                                                          

Comments: 

1. In Figure 2D, the author showed Heatmap representation of differentially abundant proteins (rows) across replicates of all three crude fractions, and there are totally 80 protein groups identified as differentially abundant by ANOVA analysis of normalized LFQ intensities. Please give all proteins pattern of all three crude fractions.

2. Based on the observed ex vivo antioxidant activity, the authors predicted the potentially responsible antioxidant proteins. The 859 proteins were filtered based on selected molecular function GO annotations, yielding 108 entries with a diverse range of molecular functions (Fig. S5). Through literature research, nine different proteins were identified as having reported antioxidant activity. Could you please provide more explanation to clarify the selection of these nine proteins?

3. Please provide a high-quality volcano plot (Fig2A-2C).

Author Response

Conclusion:

In this study, Kathrine et al. employed size exclusion chromatography to fractionate the grass juice, aiming to gain a deeper understanding of its ex vivo antioxidant properties. They combined this with quantitative bottom-up proteomics, GO-term analysis, and fraction-based enrichment to provide the most detailed characterization of the L. perenne proteome and quantitative protein partitioning in the juice and pulp fractions. Their findings open possibilities for targeted process development and optimization to improve the oxidative stability of the green juice during green biorefining, leading to a higher-quality protein product.                                                                         

We thank the reviewer for their overall positive reception of our work and are also happy that the reviewer got the main take-home message out of the work that we tried to convey.

Comments:

  1. In Figure 2D, the author showed Heatmap representation of differentially abundant proteins (rows) across replicates of all three crude fractions, and there are totally 80 protein groups identified as differentially abundant by ANOVA analysis of normalized LFQ intensities. Please give all proteins pattern of all three crude fractions.

We agree that this would be beneficial. But as dataset contains >1000 identified proteins and < 80 were differentially abundant, we decided to initially only include these as the relevant to describe the effect of the process. But to get a full overview, as we agree would be beneficial, we have included a heatmap of all proteins in the supplementary material (New Figure S3) and introduced this in the text (L516-517)

  1. Based on the observed ex vivo antioxidant activity, the authors predicted the potentially responsible antioxidant proteins. The 859 proteins were filtered based on selected molecular function GO annotations, yielding 108 entries with a diverse range of molecular functions (Fig. S5). Through literature research, nine different proteins were identified as having reported antioxidant activity. Could you please provide more explanation to clarify the selection of these nine proteins?

We fully acknowledge the comment and request from the reviewer. This comment made us revisit our analysis and perform a full walk-through of the steps. This in fact allowed us to identify an error early in the process, why a thorough re-analysis was performed. This resulted in some differences in the outcome of the analysis, which have now been included in the revised version. Briefly summarized, our initial GO-term analysis now shows additional proteins, which has been revised. Moreover, the number of proteins going through our filtering process also increased. Table 3 now includes all proteins identified through GO-term analysis and subsequent filtering of the subset based on quality of identification and abundance in the analysis of the fractionated green juice. The filtered subset was the basis for the thorough literature search and evaluation of experimental evidence for in vitro antioxidant activity and thereby the potential of the proteins showing ex vivo activity in a protein-product from refining the green juice further. The approach for GO-term analysis and subsequent filtering has now been described in more detail in both M&M (L389-414) as well as briefly introduced in R&D (L701-713). Based on this, we have revised the text related to this part of the analysis but also implemented a slight restructuring of the text, so Table 3 now comes after Figure 6, as we feel this improves the flow of the storyline. This also means that the analysis of abundance in the three initial crude fractions and Figure 7 has been revised accordingly. In fact, this comment has been the basis for a major revision of sections 3.5 to 3.7. Nevertheless, the overall outcome and conclusion does fortunately not change. Small changes have been implemented in the conclusions as well (L853-858) to reflect the change in output from the analysis.

We thank the reviewer for drawing our attention to this and are pleased to have identified the error, thereby improving the quality of the paper. We hope the revision and more detailed description of the workflow is satisfactory to the reviewer.

  1. Please provide a high-quality volcano plot (Fig2A-2C).

We fully acknowledge the concern and criticism on this point. We wanted to illustrate differential proteins between fractions from pair-wise comparison, but also see that by including three, the detail level was perhaps not satisfactory. To address this, we have decided to improve the Volcano plots (particularly in size and resolution) and only include the grass vs. juice plot in the main text (Fig. 2A), as we find this comparison the most important. What changes from the starting material to the most interesting stream for downstream processing (and the stream we subsequently fractionate). The remaining two volcano plots have similarly been improved but now moved to the supplementary (grass vs. pulp as fig. S2 and juice vs. pulp as fig. S3) to avoid overcrowding the manuscript with figures. We have therefore also changed figure references and restructured the text describing them (L499-504) a bit.

Reviewer 2 Report

Comments and Suggestions for Authors

The manuscript involves an important topic related to sustainable protein recovery from perennial ryegrass, which aligns with  global trends in green biorefining and plant-based protein research. However few issues should be addressed before it is suitable for publication.

1) The introduction should be restructured. The specific role of perennial ryegrass in biorefining should be more discussed. 

2) Figures are too dense. Its difficult to interpret anything from the volcano plots.

3) There are no experimental validation of antioxidant properties.

4) Typographical errors and grammatical inconsistencies should be looked into

Author Response

The manuscript involves an important topic related to sustainable protein recovery from perennial ryegrass, which aligns with global trends in green biorefining and plant-based protein research. However few issues should be addressed before it is suitable for publication.

We thank the reviewer for recognizing relevance, importance, and overall quality of our work and happily address the comments supplied during review.

1) The introduction should be restructured. The specific role of perennial ryegrass in biorefining should be more discussed.

We respectfully ask for more detailed information on how the reviewer thinks the introduction should be restructured to improve manuscript quality. In our opinion, the introduction is logically structured has specific focus on green biorefining and the related challenges. We initially outline the broader context and problem and then narrow in on grasses to briefly describe their advantages as a protein crop. Then we narrow further into perennial ryegrass and describe its basic characteristics. Then we describe that one way to utilize such crops is through green biorefining and describe the concept and strategies in general terms. Then we describe the main characteristics of a product and what is currently known on the protein level – leading to the challenges of oxidative reactions in the juice. This highlights why the antioxidant proteins are particularly interesting from a process POV as knowing and understanding them may facilitate process improvements. This leads directly to the knowledge gap that we want to address with our study.

If the point of the reviewer (as we interpret it) is that the application of green biorefining on perennial ryegrass specifically and not “just” grasses in general is insufficiently covered, then we acknowledge his point and agree that there are not many details on this. One of the reasons is there is not much available on perennial ryegrass as an isolated species in this context, as most studies use mixed fields. We have gone through literature once more and found a handful more works relevant to include. This has also resulted in substantial changes and additions to the text (L76-92). We hope this satisfies the reviewer – if not, please provide more detailed feedback on what you think we could improve in the introduction.

2) Figures are too dense. Its difficult to interpret anything from the volcano plots.

We fully acknowledge the concern and criticism on this point. We wanted to illustrate differential proteins between fractions from pair-wise comparison, but also see that by including three, the detail level was perhaps not satisfactory. To address this, we have decided to improve the Volcano plots (particularly in size and resolution) and only include the grass vs. juice plot in the main text (Fig. 2A), as we find this comparison the most important. What changes from the starting material to the most interesting stream for downstream processing (and the stream we subsequently fractionate). The remaining two volcano plots have similarly been improved but now moved to the supplementary (grass vs. pulp as fig. S2 and juice vs. pulp as fig. S3) to avoid overcrowding the manuscript with figures. We have therefore also changed figure references and restructured the text describing them (L499-504) a bit.

3) There are no experimental validation of antioxidant properties.

This is correct on the single-protein level. We would, however, like to add a few comments to this point. In our screening, we used a three-point dilution in our assays, showing concentration-dependent antioxidant activity in vitro. We then used the reconstituted fractions to determine an EC50 for the individual fractions, thereby reflecting that the observed activity is indeed true and not the result of false positives during screening.

While this also entails, as described in the manuscript, that our findings are indicative and requires validation on the single protein level, obtaining such validation is unfortunately beyond the scope of this work due to limited resources. It would require highly selective isolation/purification, recombinant expression/secretion/isolation, or obtaining pure proteins commercially.

In our delimitation of potentially antioxidant proteins from GO-term analysis, we enforced a requirement that proteins must have known and validated in vitro activity on the single protein level in independent studies as an inclusion criterion. In our opinion, this also serves as indirect a priori validation of their properties. During manuscript revision, we have refined this selection process, leading to a revision of the presented and discussed proteins. We have in this process also tried to emphasize the aspect of lacking validation but that independent studies have provided a priori evidence for their in vitro antioxidant activity.

While we agree that it would have been optimal to validate our findings, we are not able to provide such data. But as also described in the manuscript, this is something that we hope to dive further into in future work. Moreover, we think that such experiments would add quite a substantial amount of text and data to an already packed manuscript.

4) Typographical errors and grammatical inconsistencies should be looked into

All authors have thoroughly gone over the manuscript again to identify any such errors. We have found some, and correcting these has improved the quality of the manuscript. We find that the manuscript is of a high linguistic and grammatical quality – even more so now. While the identified errors naturally improved the manuscript, we urge the reviewer to pinpoint any specific errors (even if systematical), that they may have observed during review.